# ZIF-67/SA@PVDF Ultrafiltration Membrane with Simultaneous Adsorption and Catalytic Oxidation for Dyes

Kaixuan Zhu [1], Shabin Mohammed [2] , Hai Tang [1,*], Zongli Xie [3] , Sisheng Fang [1] and Shasha Liu [1,*]

1   School of Chemical and Environmental Engineering, Anhui Polytechnic University, Wuhu 241000, China
2   NYUAD Water Research Center, Department of Engineering, New York University Abu Dhabi, Abu Dhabi P.O. Box 129188, United Arab Emirates
3   CSIRO Manufacturing, Private Bag 10, Clayton South, VIC 3169, Australia
*   Correspondence: tanghai@ahpu.edu.cn (H.T.); liushasha@ahpu.edu.cn (S.L.)

**Abstract:** Due to their stable chemical properties and complex structures, dyes are difficult to be removed from water. Herein, a ZIF-67/SA@PVDF (ZSA3@PVDF) mixed matrix membrane has been fabricated by incorporating silicon aerogel (SA) and zeolitic imidazolate framework material 67 (ZIF-67) nanoparticles in a polyvinylidene fluoride (PVDF) membrane for the removal and degradation of dyes from water. The influence of SA and ZIF-67 on the morphology and structure of the membrane was confirmed using scanning electron microscope (SEM) and atomic force microscope (AFM). In ZSA3@PVDF membrane, both SA and ZIF-67 are highly porous nanomaterials that possess good adsorption capacity, as confirmed by the Brunauer–Emmett–Teller (BET) result. In addition, the cobalt (Co) element of ZIF-67 can catalyze peroxymonosulfate (PMS) to generate strong oxidizing sulfate radicals ($SO_4^{2-}$), contributing to improving regeneration capacity of the ZIF-67/SA@PVDF membrane. The water flux of ZSA3@PVDF membrane is 427.6 L m$^{-2}$ h$^{-1}$ bar$^{-1}$, and the Methylene blue (MB) removal rate is higher than 99% when filtrating 100 mL MB solution (5 mg/L). The regeneration test result shows that the removal rate of the ZSA3@PVDF membrane is still above 98% after five cycles of adsorption of MB. The self-cleaning experiment shows that the adsorption of SA in the ZSA3@PVDF membrane promotes the catalytic performance of the membrane, showing a better self-cleaning ability. The ZSA3@PVDF membrane provides a new strategy for the removal of dyes in the advanced purification of dye wastewater.

**Keywords:** dye; silicon aerogel; metal-organic frameworks; adsorption; mixed matrix membrane

## 1. Introduction

Dyes are widely used in the leather, plastic, textile, food processing, paper, printing and dyeing industries [1]. Water pollution caused by excessive discharge of dyes has obtained considerable attention due to serious health and environmental impact [2]. For most dye pollutants, aromatic rings with carcinogenic risk usually exist in the molecular structure, posing a serious threat to public health and ecological security [3]. Therefore, these dye pollutants must be monitored and removed from the environment to reduce their impacts [4]. Removing dye pollutants before discharge of wastewater is an effective strategy, which is conducive to improving the reuse rate of wastewater, alleviating water resources, and promoting the sustainable development of the environment [5].

Many methods have been developed to treat the organic pollution in the water, including adsorption, photocatalysis, biological treatment, advanced oxidation process (AOP) and membrane filtration [6–10]. Among them, membrane and advanced oxidation water treatment technologies have gained increasing acceptance [11,12]. Membrane separation technology has been widely used in water pollution treatment due to its characteristics of selective separation, strong adaptability, easy maintenance, small floor area, and high efficiency [13]. Advanced oxidation is another effective pollutant remediation method. Among

various advanced oxidation methods, catalytic oxidation of peroxymonosulfate (PMS) has attracted much attention due to its wide operating pH range (2.0–8.0) and high standard oxidation potential (up to 3.1V) [14]. The catalytic membrane coupled with advanced oxidation processes (AOPs) not only significantly enhances the pollutant removal efficiency but also inhibits the fouling of the membrane via self-cleaning [15,16]. Nevertheless, several drawbacks still exist during PMS activation. Unnecessary agglomeration usually occurs between nano catalysts, resulting in limited available active centers and low mass transfer efficiency [17]. In addition, most catalysts are tough to recover and reutilize [18]. Dispersing nano catalysts in the membrane matrix is an effective way to solve this problem.

Metal-organic frameworks (MOFs) and their derivatives show potential applications in the field of PMS activation due to their unique physicochemical properties, including ultra-high surface area, regular and highly controlled porosity [19–22]. Zeolitic imidazolate frameworks (ZIFs) are an attractive subclass of MOFs. Compared with most other MOFs, ZIFs have high porosity, large surface area, ease of synthesis, easily available coordination unsaturated sites, and excellent chemical and solvent stability [23,24]. In particular, ZIF-67, a ZIF composed of cobalt ions and 2-methylimidazolium salts, has become a promising adsorption technology material [25,26]. Ye et al. reported a ZIF-67/CNTs-II@PVDF membrane with excellent catalytic property and good reusability [27]. Du et al. conducted N-doped CNTs with embedded Co nanoparticles from ZIF-67 for PMS activation to degrade bisphenol A [28]. Yin et al. reported a SPSf membrane fabricated via a nonsolvent-induced phase separation method using ZIF-67 as a crosslinker, which maintained high thermal stability and better antifouling performance (FRR = 70%) [29]. In our previous work, we reported a ZIF-67@PVDF membrane for dye wastewater treatment via sulfate radical enhancement [30]. The adsorbed dye (AO7) is degraded into small molecules in the presence of the reactive oxygen species (ROS) produced from the highly efficient PMS-activated ZIF-67@PVDF membrane.

Adsorption technology can be employed for facile and efficient removal of pollutants through transfer of pollutants from one phase to another. The pollutants adsorption on catalysts has been identified as a critical step for efficient catalytic oxidation [31]. The pollutants degradation is effectively promoted by adsorption, resulting in a further increase in adsorption [32]. However, the role of pollutant adsorption in the oxidation process has been excluded or ignored in many studies. Considering the disadvantages/advantages of adsorption and PMS-AOPs technology, it should be a feasible strategy to prepare bifunctional materials for efficient dyes' removal to make them complementary. $SiO_2$ aerogel (SA) has a three-dimensional porous network structure with numerous air-filled pores. Due to its high porosity, extremely low density, and high surface aera, SA has gradually replaced $SiO_2$ nanoparticles and been used as the adsorbent for harmful substances [33]. Ruan et al. obtained a novel composite aerogel of self-assembled cellulose nanocrystals (CNCs)/$SiO_2$ by freeze-drying in order to adsorb methylene blue (MB) pollutants in water [34]. The combination of adsorption and catalytic membrane will be a promising research direction. To the best of our knowledge, no prior research has been conducted on the use of SA adsorption in catalytic membranes.

In this study, we report a simple and effective method to fabricate ZIF-67/SA@PVDF (ZSA3@PVDF) multifunctional membrane with simultaneous ultrafiltration, adsorption and catalytic oxidation performance. The porous SA and ZIF-67 nanoparticles were incorporated in PVDF membrane. The characteristics of the SA and ZIF-67 nanoparticles were analyzed. Membranes containing different amounts of SA (up to 3% by weight) were systematically studied to investigate the effect of various parameters, including solution pH, temperature and initial concentration on the adsorption performance. In addition, the self-cleaning performance of the SA3@PVDF, ZIF-67@PVDF and ZSA3@PVDF membranes were investigated. Moreover, the mechanism of multifunctional decontamination in the ZSA3@PVDF membrane was studied systematically. This study provides a new strategy for a multifunctional UF membrane with simultaneous adsorption and catalytic oxidation.

## 2. Experimental Work

### 2.1. Materials

PVDF powder (SALAVY 6020). Peroxymonosulfate (PMS, 47%), methylene blue (MB), rhodamine B (RhB), and orange yellow (AO7) were purchased from Shanghai Aladdin Biochemical Co., Ltd. (Shanghai, China) Silicon aerogel powder (SA) was supplied by Anhui Keang Nano Technology Co., Ltd. (Chuzhou, China) N-N-2 methyl acetamide (DMAc, 99%, AR), cobalt nitrate hexahydrate (Co(NO$_3$)$_2$·6H$_2$O, 99.99%), and 2-methylimidazole (2-Melm, 98%) were purchased from Shanghai Aladdin Biochemical Co., Ltd.

### 2.2. Preparation of ZIF-67

In a typical synthesis, 0.52 g cobalt nitrate hexahydrate was dissolved in 20 mL of deionized (DI) water; then, 0.9852 g 2-methylimidazole was dissolved in 20 mL of DI water. Those two solutions were mixed, stirred for 30 min, and left at room temperature for 12 h. Then, the resulting purple precipitates were collected by centrifuging, washed with water and methanol for 3 times, and vacuum-dried at 60 °C for 24 h.

### 2.3. Preparation of PVDF, ZIF-67@PVDF and (Z)SAx@PVDF Membranes

The mixed matrix membranes were prepared using a phase inversion method as reported elsewhere [35]. The casting solutions were prepared according to the components (including PVDF, ZIF-67, SA, and DMAc) summarized in Table 1, and the main membrane fabrication process is illustrated in Figure A1. The casting solutions were prepared by dissolving them in DMAc at 40 °C. After stirring for 12 h, the homogeneous casting solutions were placed in a vacuum-drying oven at 60 °C for 4 h to eliminate air bubbles. Subsequently, the solution was casted on a glass plate using a casting knife, followed by immersion of the membrane into DI water to induce phase inversion. The membranes with different SA content are named (Z)SAx@PVDF, where x value (x = 1, 2, 3) indicates the concentration of SA in the casting solution.

**Table 1.** The composition of different casting solution used for membrane fabrication.

| Membranes | SA (wt%) | PVDF (wt%) | ZIF-67 (wt%) | DMAC (wt%) |
|---|---|---|---|---|
| PVDF | 0 | 8 | 0 | 92 |
| ZIF-67@PVDF | 0 | 8 | 1 | 91 |
| SA1@PVDF | 1 | 8 | 0 | 91 |
| SA2@PVDF | 2 | 8 | 0 | 90 |
| SA3@PVDF | 3 | 8 | 0 | 89 |
| ZSA3@PVDF | 3 | 8 | 1 | 88 |

### 2.4. Characterizations

The specific surface area and average pore diameter of SA powder and ZIF-67 nanoparticles were characterized by the Brunauer–Emmett–Teller (BET)-ASAP2460 analyzer.

The surface and cross-section morphologies of the prepared membrane samples were observed using a field emission scanning electron microscope (FE-SEM, S-4800, Hitachi, Tokyo, Japan). At least three measurements of each sample were tested to ensure the reliability of results. The surface morphology and roughness of the as-prepared membranes were analyzed with an atomic force microscope (AFM, Bruker Dimension Edge). Sample dimension was kept uniform, 3 µm × 3 µm with the scanning area measuring 2 µm × 2 µm. The average roughness (*Ra*) and root mean square roughness (*Rq*) were evaluated to characterize the membrane surface roughness, which was quantitatively calculated by a software (NanoScope Analysis). The membrane porosity was measured by testing both the wet and dry weights. The wet membrane samples were dried in a vacuum oven at 60 °C for 24 h before measuring the dry weight.

## 2.5. Evaluation of Adsorption Performance

The adsorption experiments were performed according to the following procedures: Static adsorption: a square sample (3 cm × 3 cm) was immersed into a 100 mL organic dye solution with a predetermined concentration and a specific pH value (preadjusted using 0.1 M HCl or 0.1 M NaOH) and stirred for 12 h. The adsorption rate of dyes was calculated based on the concentration of the solution collected at different time intervals, which was determined through UV–vis spectroscopy using a spectrophotometer. The adsorption capacity ($q$, mg g$^{-1}$) and the removal efficiency ($R$, %) are calculated according to Equations (1) and (2):

$$q = \frac{(C_0 - C_1) \times V}{S} \tag{1}$$

$$R = \frac{(C_0 - C_1)}{C_0} \times 100\% \tag{2}$$

Here, $C_0$ and $C_1$ (mg/L) are the initial and final concentrations of the contaminant material, $V$ (L) is the volume of the contaminant solution, and $S$ (cm$^2$) is the area of the membrane.

The dynamic adsorption performance of the membranes was tested using a cross-flow equipment. The effective membrane surface area was 25.12 cm$^2$. Prior to testing, all membranes were pressurized at 0.15 MPa for 30 min to minimize the effect of compaction. The pure water flux of the membrane ($J_{wv}$) was obtained by filtering the deionized water at 0.1 MPa for 30 min, whereas the flux of dye solution ($J_{wc}$) was determined by subsequently filtering 200 mL MB solution (5 mg/L). The weight of the filtrated solution was weighed every 2 min with an electronic scale until steady state values were obtained. The flux of the membrane was calculated by Equation (3).

$$J = \frac{m}{\rho A \Delta t} \tag{3}$$

where $J$ is the permeate flux (L/m$^2$h), $m$ is the permeate weight (kg), $A$ the membrane area (m$^2$), $\rho$ is the density of permeate water (1.0 kg/L), and $\Delta t$ is the filtration time (h).

## 2.6. Evaluation of Self-Cleaning Performance

The self-cleaning performance of the prepared membranes was conducted. PVDF, SA3@PVDF, ZIF-67@PVDF and ZSA3@PVDF membrane sample were put in 50 mL MB solution (20 mg/L) for 2 h to reach adsorption saturation. Then, each saturated membrane was added into 50 mL solution (containing 20 mg/L MB and 1 g/L PMS). After 1 h, the solution concentration was measured with an UV spectrophotometer. This MB + PMS cleaning step was repeated three times.

## 3. Result and Discussion

### 3.1. Characterizations

Figure 1 presents the SEM images of SA (Figure 1a) and ZIF-67 (Figure 1b), revealing their morphology. The SA comprised a highly porous interconnected 3D network structure, whilst the ZIF-67 nanoparticle had a representative rhombic dodecahedral appearance with a mean particle size in the range of 300–600 nm (Figure 1c). The FTIR spectra of ZIF-67 (Figure 1d) show the Co–N bond stretching vibration occurred at 429 cm$^{-1}$, confirming the successful preparation of ZIF-67. The absorption band at 752 cm$^{-1}$ is assigned to C=N stretching vibration, while the band at 1301 cm$^{-1}$ corresponds to C=C stretching. The band at 1421 cm$^{-1}$ is attributed to CH$_3$ bending vibration. The band at 1640 cm$^{-1}$ is assigned to N-H bending. In addition, with the peak at 993 cm$^{-1}$, the existence of C-N stretching vibration is proved by the appearance of a band at 1140 cm$^{-1}$ [36].

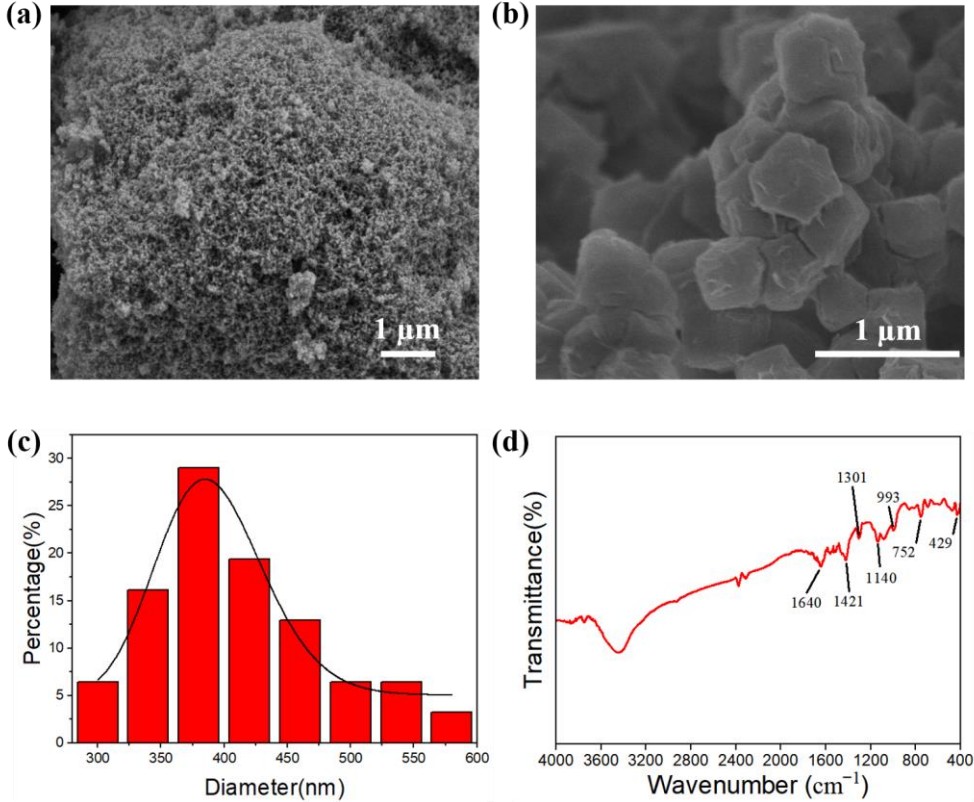

**Figure 1.** SEM images of material surface (**a**) SA, (**b**) ZIF-67. (**c**) article size distribution and (**d**) FTIR spectra of ZIF-67.

The specific surface area and average pore diameter of SA and ZIF-67 nanoparticles are presented in Table 2. BET is considered a common approach to assess the specific surface area [37]. The specific surface area of SA and ZIF-67 nanoparticles was estimated to be 887.39 and 1363.94 $m^2/g$, respectively. Meanwhile, the average internal pore diameter of SA powder and ZIF-67 nanoparticles were around 3.66 and 2.61 nm, respectively. The BET results indicate that both SA powder and ZIF-67 nanoparticles are highly porous in nature, which is desirable for adsorption materials.

**Table 2.** Specific surface area, and average pore diameter of SA and ZIF-67.

| Material | Specific Surface Area ($m^2/g$) | Average Pore Diameter (nm) |
|---|---|---|
| SA | 887.39 | 3.66 |
| ZIF-67 | 1363.94 | 2.61 |

The surface morphology of PVDF, ZIF-67@PVDF and (Z)SAx@PVDF membranes is shown in Figure 2. The surface of the pristine PVDF membrane had many macro voids in the range of several hundred nanometers (Figure 2a). With the ZIF-67 incorporation, the membrane exhibited smaller surface pores, as revealed in the SEM image of the ZIF-67@PVDF membrane (Figure 2b). With the addition of SA in the PVDF membrane (Figure 2c–e), a relatively denser surface morphology was observed with smaller holes, and the membrane surface became rougher, which is consistent with the AFM result which will be discussed in the subsequent sections. However, compared with the SA3@PVDF membrane, the pore size of the ZSA3@PVDF membrane (Figure 2f) was larger; this may be due to the interaction between silica aerogel and ZIF-67 nanoparticles.

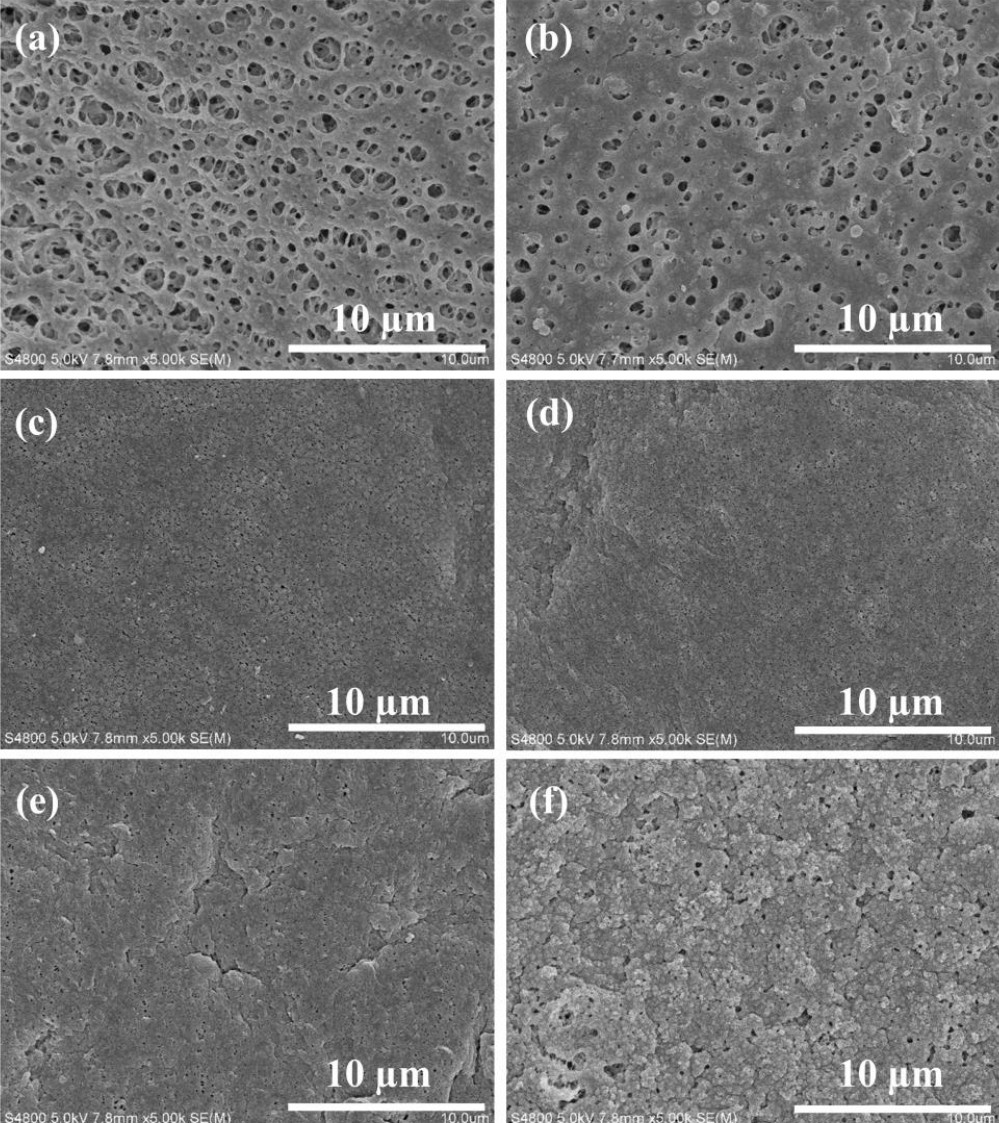

**Figure 2.** SEM images of membrane surface (**a**) PVDF membrane, (**b**) ZIF-67@PVDF membrane, (**c**) SA1@PVDF membrane, (**d**) SA2@PVDF membrane, (**e**) SA3@PVDF membrane, and (**f**) ZSA3@PVDF membrane.

The cross-sectional morphologies of all membrane samples were also observed to further evaluate the effect of SA and ZIF-67 on the PVDF membrane. As presented in Figure 3, the addition of SA makes the PVDF membrane more compact. For instance, compared with SA1@PVDF membrane, the cross-sections of the SA2@PVDF and SA3@PVDF membranes were denser with a transition from the finger-like large voids near the top surface to denser sponge structure. As seen in Figure 3f, the ZSA3@PVDF membrane showed a relatively looser structure than the SA3@PVDF membrane, which is probably due to the hydrophilic ZIF-67 addition. The SEM cross-section result demonstrates that the SA makes the PVDF membrane denser, and the ZIF-67 addition will enlarge the pore size of the PVDF or SA3@PVDF membrane.

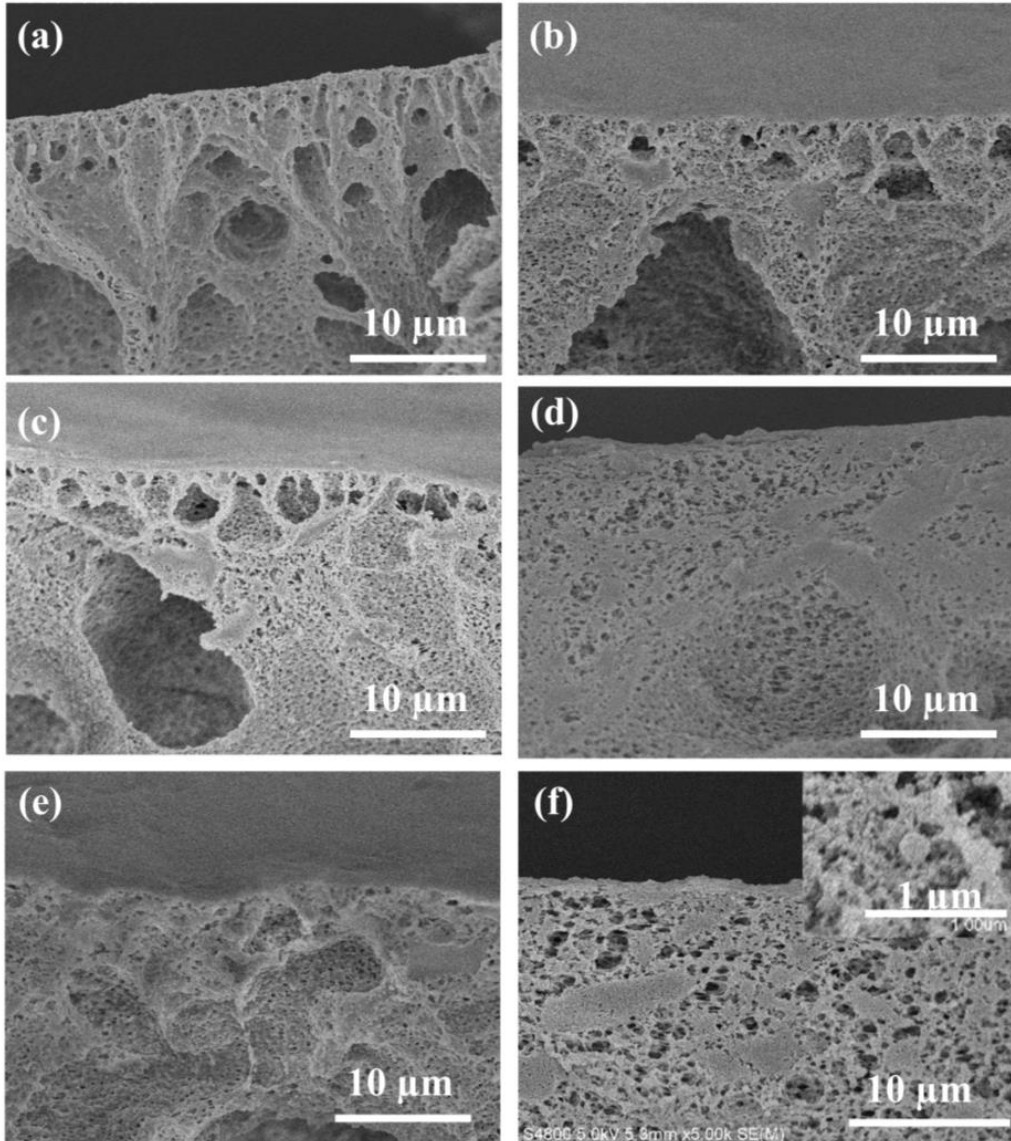

**Figure 3.** Cross-section SEM images of (**a**) PVDF membrane, (**b**) ZIF-67@PVDF membrane, (**c**) SA1@PVDF membrane, (**d**) SA2@PVDF membrane, (**e**) SA3@PVDF membrane, and (**f**) ZSA3@PVDF membrane.

Surface properties such as surface roughness have significant influence on the properties of the UF membrane. The 3D surface morphology and roughness of the PVDF, ZIF-67@PVDF and (Z)SAx@PVDF membrane were measured with AFM (Figure 4). From the changes in the roughness parameters (Ra and Rq) listed in Table 3, it can be seen that the addition of SA or ZIF-67 had a considerable effect on the surface roughness of the prepared membranes. Membrane roughness increased considerably with the addition of SA or ZIF-67 nanoparticles. We also observed a gradual increase in the roughness of the SAx@PVDF membrane, from 15.4 nm to 94.3 nm, with the addition of SA. Furthermore, the highest roughness was observed for the ZSA3@PVDF membrane (112 nm), which was considerably higher than that of the SA3@PVDF membrane, which could be attributed to the ZIF-67 nanoparticles.

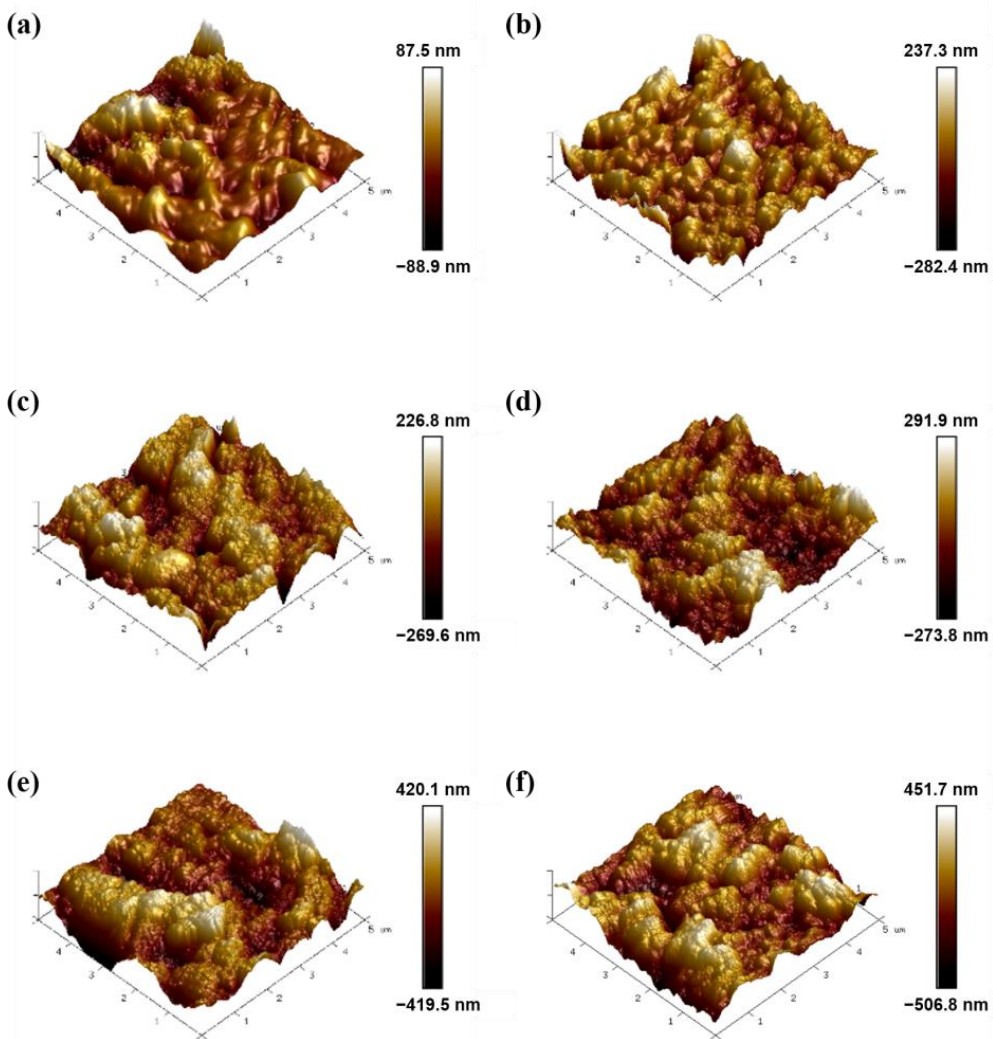

**Figure 4.** AFM images of (**a**) PVDF membrane, (**b**) ZIF-67@PVDF membrane, (**c**) SA1@PVDF membrane, (**d**) SA2@PVDF membrane, (**e**) SA3@PVDF membrane, and (**f**) ZSA3@PVDF membrane.

**Table 3.** The roughness of the PVDF, ZIF-67@PVDF and (Z)SAx@PVDF membrane.

| Membranes | Ra (nm) | Rq (nm) |
|-----------|---------|---------|
| PVDF | 15.7 | 21.7 |
| ZIF-67@PVDF | 47 | 64 |
| SA1@PVDF | 52.8 | 67.4 |
| SA2@PVDF | 65.8 | 82.3 |
| SA3@PVDF | 94.3 | 83.6 |
| ZSA3@PVDF | 112 | 139 |

### 3.2. Static Adsorption Performance

The effects of pH, adsorption time, and initial concentration on the adsorption performance of the ZSA3@PVDF membrane were studied with MB. The adsorption efficiency of MB at different adsorption times is shown in Figure 5a. Compared to the pristine PVDF membrane, the adsorption of MB dye for the mixed matrix membranes was fast for the initial 120 min and then gradually slowed until the equilibrium was reached at around 480 min. As displayed in Figure 5a, the adsorption capacity of the SAx@PVDF membrane was significantly improved compared with that of the pristine PVDF membrane. In addition to this, with the increase in SA addition, the static adsorption capacity of the SA@PVDF membrane is significantly enhanced. The adsorption saturation of the ZSA3@PVDF mem-

brane was reached after 8 h and was the highest among all the prepared membranes, which is about 140% higher than the pristine PVDF membrane. The adsorption of organic dyes in aerogels is affected by hydrogen bonding, electrostatic interactions, and number of the absorption sites [38,39]. Accordingly, SA3@PVDF and ZSA3@PVDF contain more adsorption sites that perform better for the dye adsorption. It is worth mentioning that the addition of the porous SA and ZIF-67 nanoparticles enhanced the adsorption capacity of the ZSA3@PVDF membrane significantly compared with that of PVDF.

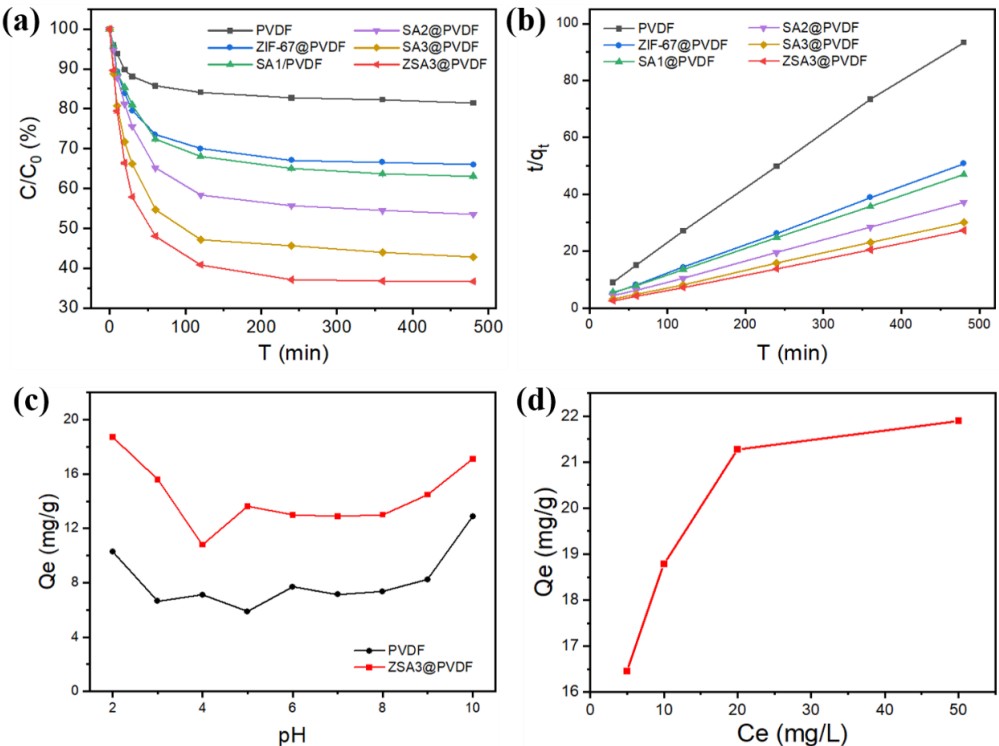

**Figure 5.** (**a**) Static adsorption capacity of MB for the PVDF, ZIF-67@PVDF and (Z)SAx@PVDF membrane. (**b**) Pseudo-second-order plot of MB on the PVDF, ZIF-67@PVDF and (Z)SAx@PVDF membrane. (**c**) Effect of solution pH on MB adsorption capacity for the PVDF and ZSA3@PVDF membrane. (**d**) Effect of initial MB concentration on adsorption capacity for the ZSA3@PVDF membrane.

The Langmuir model and the Freundlich models are commonly used models to describe the adsorption isotherms. Langmuir adsorption isothermal model is used to describe single-molecule adsorption on homogeneous surfaces, and a Freundlich adsorption isothermal model is used to analyze multilayer adsorption on inhomogeneous surfaces. The Langmuir equation and the Freundlich equations are represented by:

$$q_t = q_e \left(1 - e^{-k_1 t}\right) \tag{4}$$

$$q_t = \frac{k_2 q_e^2}{1 + k_2 q_e t} \tag{5}$$

where $q_e$ is the equilibrium adsorption capacity of adsorbate, $q_t$ (mg g$^{-1}$) is the equilibrium adsorption capacity at time $t$ (min); and $k_1$ (min$^{-1}$) and $k_2$ (mg g$^{-1}$min$^{-1}$) are the rate constant of pseudo-first-order and pseudo-second-order kinetics, respectively.

The typical adsorption kinetic models were applied to fit the adsorption data. These parameters in Figure 5b were calculated and summarized in Table 4. As a result, the pseudo-second-order model was found to be the most suitable to describe the adsorption of MB on these prepared membranes considering its high correlation coefficient ($R^2 > 0.996$,

Table 4). This result indicates that the adsorption of MB on the prepared membranes is mainly controlled by a chemical process, and affected by membrane performance and MB structure. It can be concluded that MB adsorption includes two steps. Firstly, the MB molecule enters the membrane pore in one direction; secondly, the π-π and p-π forces occur between the membrane and MB molecules. The second step controls the rate of the adsorption process.

**Table 4.** Kinetic parameters for MB onto PVDF membranes adsorbent.

| Membranes | Pseudo-First-Order | | | Pseudo-Second-Order | | |
|---|---|---|---|---|---|---|
| | $k_1$ (min$^{-1}$) | $R_1^2$ | $q_e$ (mg g$^{-1}$) | $k_2$ (min$^{-1}$) | $R_2^2$ | $q_e$ (mg g$^{-1}$) |
| PVDF | 0.0413 | 0.9827 | 4.77 | 0.0104 | 0.9974 | 5.22 |
| ZIF-67@PVDF | 0.0333 | 0.9900 | 9.05 | 0.0044 | 0.9993 | 9.96 |
| SA1@PVDF | 0.0268 | 0.9906 | 9.84 | 0.0030 | 0.9972 | 10.98 |
| SA2@PVDF | 0.0267 | 0.9952 | 12.48 | 0.0024 | 0.9971 | 13.90 |
| SA3@PVDF | 0.0356 | 0.9848 | 15.21 | 0.0029 | 0.9960 | 16.60 |
| ZSA3@PVDF | 0.0419 | 0.9908 | 17.14 | 0.0028 | 0.9971 | 18.68 |

The effect of solution pH on MB removal was further investigated as the pH might influence the surface charge of the adsorbent membrane. The batch experiments under various initial pH values (2.0–10.0) were conducted in a 5 ppm MB solution with a 3 cm × 3 cm membrane for 10 h. As displayed in Figure 5c, when the pH value increased from 2.0 to 10.0, we initially observed a sharp decrease in the adsorption capacity, which was followed by a noticeable increase upon further increase in pH, revealing that the membrane is more effective for dye adsorption under strong acid or alkali conditions. ZSA3@PVDF shared a similar trend with the PVDF membranes under different pH conditions. The silane phenolic groups and matrix oxygen atomic groups on the SA surface can form hydrogen bonds with the dye at an appropriate concentration, thereby promoting dye adsorption. At pH = 10, the MB adsorption trend of ZSA3@PVDF can be inhibited, which is mainly due to the negative charge of MB in alkaline solution, and the electrostatic exclusion between SA particles. These indicate that the adsorption membrane has good adsorption performance under a wide range of acidic and alkaline conditions.

The adsorption capacity of MB on the prepared ZSA3@PVDF membrane was also estimated under different initial concentrations of adsorbent solution in the range of 5–50 ppm. All the adsorption results reported here were obtained by conducting experiments with a contact time of 12 h to ensure the adsorption equilibrium has attained. As shown in Figure 5d, at higher initial concentration of MB solution, a visible increase in the equilibrium adsorption capacity ($q_e$) was observed. The adsorption achieved the saturation state when the initial concentration of MB reached 50 mg/L, suggesting that all the potential active sites were completely occupied. This may be due to the increase in active adsorption sites by adding SA with a porous network structure.

Further, the performance of membranes as regards the adsorption of different dyes was evaluated by conducting experiments with AO7 and Rhodamine B (Table 5). The adsorption capacity of the ZSA3@PVDF membrane was found to be higher than that of the PVDF membrane irrespective of the dyes tested. Notably, the adsorption efficiency of the AO7 adsorption on the ZSA3@PVDF membrane was observed to be much higher than that of the PVDF membrane, possibly due to the charge effect. For instance, both RhB and MB are cationic dyes, while AO7 is an anionic dye. This would result in an electrostatic attraction between SA and the dye, which promotes the adsorption of AO7.

**Table 5.** Absorption ability of membranes for different dyes (5 ppm).

| Membranes | Adsorption Capacity for Different Dyes | | |
| --- | --- | --- | --- |
| | AO7 | RhB | MB |
| PVDF | 2.505 | 17.11 | 5.22 |
| ZSA3@PVDF | 11.875 | 21.165 | 18.68 |

### 3.3. The Permeability and Dynamic Removal Ability of MB

As displayed in Figure 6a, with the SA or ZIF-67 nanoparticles addition, the flux of mixed matrix membrane may decrease considerably due to the decrease in membrane pore size (as seen in Figure 2). For example, the flux of the membranes decreased from 632.5 L m$^{-2}$ h$^{-1}$ bar$^{-1}$ (PVDF membrane) to 335.8 L m$^{-2}$ h$^{-1}$ bar$^{-1}$ (SA3@PVDF membrane) with the increase in SA proportion. This result is due to the effect of SA on membrane pore structure. The increase in SA addition, which accumulates on the membrane surface, reduced the pore size distribution on the membrane surface and caused a decrease in flux. The ZSA3@PVDF membrane somewhat restored the water flux to 427.6 L m$^{-2}$ h$^{-1}$ bar$^{-1}$ due to the presence of ZIF-67 nanoparticles, which prevented the SA from clustering to each other and simultaneously allowed the water molecules to pass through the pores of the ZIF-67. Overall, the addition of ZIF-67 and SA will make the flux reduction in MB smaller, which is related to SA capturing MB molecules, partly reducing its blockage in the membrane pore.

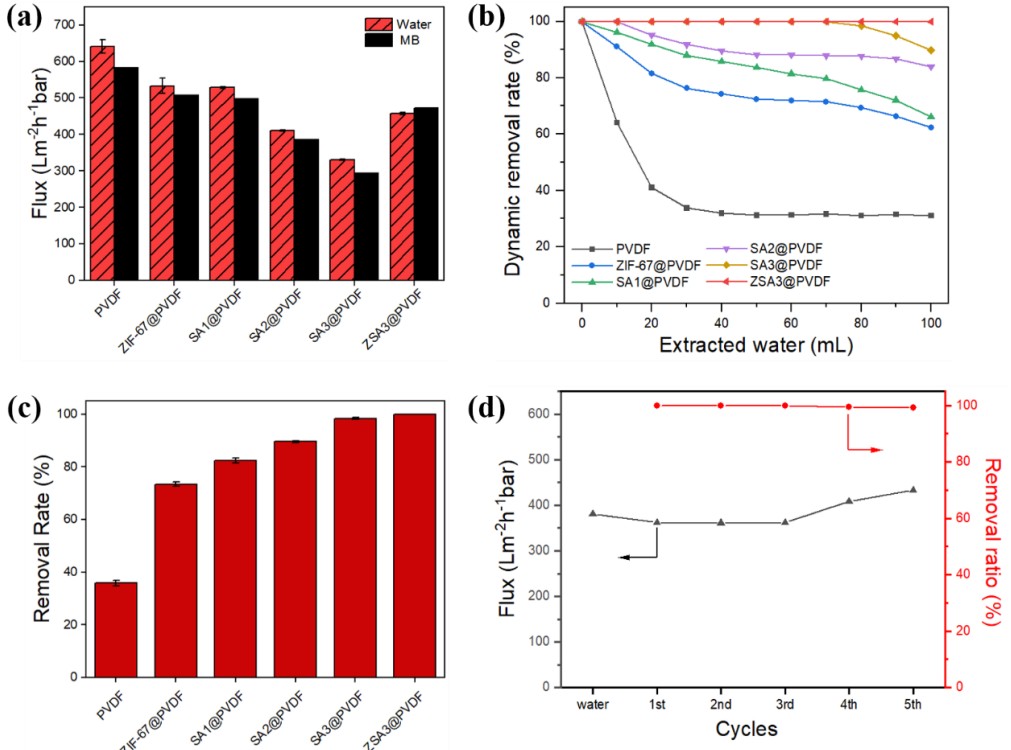

**Figure 6.** (**a**) The pure water and MB flux of different types of membrane. (**b**) The dynamic removal efficiency of different types of the membrane towards MB under various permeated volumes. (**c**) MB removal rate of different types of membrane. (**d**) Flux and cycling performance of ZSA3@PVDF membrane towards MB.

The dynamic adsorption properties of the prepared PVDF, ZIF-67@PVDF and (Z)SAx-@PVDF composite membrane were studied. As displayed in Figure 6b, in most cases, the removal efficiency was constantly decreasing with the increase in volume of the filtrated MB solution. However, compared with the original PVDF membrane, the dynamic removal efficiency of the composite membranes (containing SA or ZIF-67) has been greatly improved. The dynamic removal efficiency of the ZSA3@PVDF membrane remained higher than 99% even after 100 mL MB solution filtrated through the membrane, indicating that the ZSA3@PVDF membrane has an excellent removal efficiency towards MB.

The dynamic removal rate for MB is shown in Figure 6c. The dynamic removal efficiency of the membrane towards MB gradually increased with the addition of SA in the membrane. The dynamic removal efficiency of the ZSA3@PVDF membrane was nearly 60% higher than the pristine PVDF membrane. The reported results show that the composite membranes have a higher removal efficiency compared to the original PVDF membrane, which is due to a higher number of adsorption sites in the presence of SA and ZIF-67 with larger specific surface area. Particularly, the ZSA3@PVDF membrane showed excellent dynamic adsorption properties, with over 99% removal of MB even when the filtration volume reached 100 mL. This exceptional adsorption performance indicates that the ZSA3@PVDF membrane is promising for practical applications to treat wastewater organic pollutants.

Considering the performance, the regeneration capabilities of ZSA3@PVDF membrane were assessed by evaluating the cyclic removal efficiency and flux change for MB solution. The catalytic degradation of MB was assessed using PMS solution to clean the full adsorbed membrane. As shown in Figure 6d, the removal rate of MB in the first round was 100% and remained above 98% up to the fifth dynamic adsorption process. Although there was a slight decrease in the removal rate in the subsequent cycles, the obtained results indicate good reuse of the ZSA3@PVDF membrane in removing MB in aqueous solution. The slight decrease in the removal efficiency is mainly due to the increment in the membrane flux, resulting in a lower residence time for the solution in the membrane, which consequently reduced the adsorption. This can be explained by the fact that the Co element from ZIF-67 of the ZSA3@PVDF membrane would activate PMS to produce $SO_4^{\bullet 2-}$ and $\cdot OH$, which would attack MB dye through a series of electron transfers, hydroxylation, and hydrogen extraction reactions [40]. MB molecules were then degraded into small-molecule organic compounds on the membrane surface, and pores, mesopores and micropores present in membranes improved the mass transfer of reactants, such as PMS or pollutants, by reacting with the active sites inside the catalyst. This enhanced the adsorption and catalytic efficiency of PMS activation, consequently increasing the permeate flux and antifouling performance and alleviating concentration polarization during filtration. Generally, the ZSA3@PVDF membrane shows excellent regeneration property.

### 3.4. Membrane Self-Cleaning Performance

The self-cleaning performance of the membranes was further studied (Figure 7). The results showed that the PVDF membrane (90.6%) and the SA3@PVDF (90.7%) membrane have similar removal efficiency, while the ZIF-67@PVDF membrane (94.8%) and ZSA3@PVDF (96.7%) show higher MB removal capacity, especially the ZSA3@PVDF membrane. The SA itself does not affect the degradation of MB but, after adding ZIF-67 + SA, the ZSA3@PVDF membrane's catalytic efficiency increased by 2% compared with ZIF-67@PVDF membrane. This indicated that the adsorption of SA played a certain role in promoting the catalytic oxidation of MB by PMS. The mechanism may be that $Co^{2+}$ activated PMS to produce $SO_4^{2-}$ oxidized MB, and the concentration of MB around the reaction site decreased. At the same time, SA with good adsorption performance enriched MB on the reaction active site, which improved the reaction utilization efficiency, thus increasing the catalytic oxidation rate. Due to the synergistic effect of SA adsorption, the membrane has played a significant role in promoting the removal rate of MB, and its maximum removal efficiency is 98.3%. The catalytic activation of PMS for the degradation of MB was conducted for three consecutive

cycles by recycling the membrane as the catalyst. In general, the result showed a better self-cleaning effect.

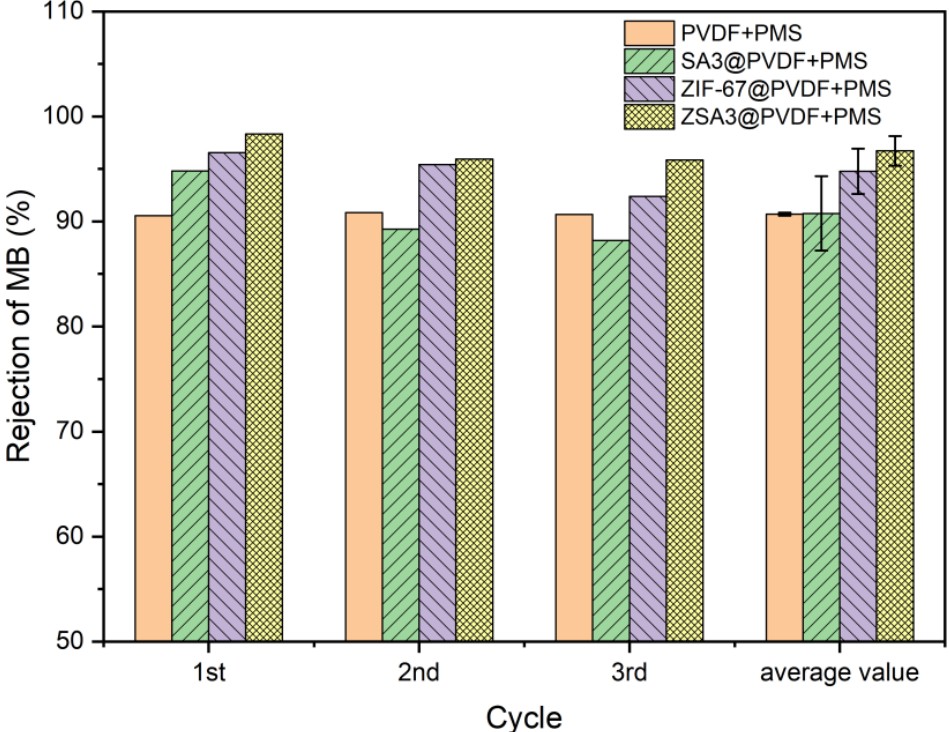

**Figure 7.** Rejection of MB (20 mg/L) after PMS cleaning.

As shown in Figure 7, the high removal efficiency was sustained, which was 96.7% for MB. This confirms the potential of using this ZSA-incorporated MMM for dye wastewater treatment. A comparison with other PVDF-based membranes for dye removal is presented in Table 6. In this work, the PMS on-line clean-assisted ZSA@PVDF ultrafiltration membrane exhibited a comparable or higher removal efficiency of dyes.

**Table 6.** Comparison with other PVDF-based membranes in removal efficiency of MB.

| Membranes | Dyes | Removal Efficiency (%) | Flux (L m$^{-2}$ h$^{-1}$ bar$^{-1}$) | Ref. |
|---|---|---|---|---|
| MIL-68(Al)/PVDF | MB | 96.30 | 41.2 | [41] |
| SDS–GO/TiO$_2$/PVDF | MB | 92.76 | 9.3 | [42] |
| GO/ZnO/PVDF | MB | 86.84 | - | [43] |
| PVDF/nanoclay/chitosan | MB | 75 | 500.0 | [44] |
| ZSA3@PVDF | MB | 98.50 | 426.0 | This work |

## 4. Conclusions

In this study, a bifunctional adsorption/catalytic UF membrane was developed. The ZSA3@PVDF membrane has a homogeneous sponge structure, and the adsorption capacity of the membrane increases with the addition of SA and ZIF-67 nanoparticles. The ZSA3@PVDF membrane showed excellent dynamic adsorption properties, with over 99% removal of MB when the filtration volume reached 100 mL. Moreover, due to the fact that ZIF-67 can activate PMS to degrade dyes, the regeneration capacity of the ZSA@PVDF membrane is obviously strengthened, and the removal efficiency is reduced from 100% to 98% after five adsorption cycles. In the self-cleaning test, the ZSA3@PVDF membrane shows better self-cleaning ability compared with ZIF-67@PVDF membrane because more dye pollutants may be concentrated by SA and then degraded. The excellent adsorption,

catalytic and filtration performance indicate that the ZSA3@PVDF membrane is promising for treating wastewater organic pollutants.

**Author Contributions:** Conceptualization, K.Z. and H.T.; methodology, K.Z. and S.M. investigation, K.Z.; data curation, S.F.; writing—original draft preparation, K.Z.; writing—review and editing, S.M. and Z.X.; supervision, S.L.; project administration, S.L.; funding acquisition, H.T. and S.L. All authors have read and agreed to the published version of the manuscript.

**Funding:** This work is financially supported by the Natural Science Foundation of Anhui Province (NO. 2108085ME188), and the Anhui Polytechnic University Startup Foundation for Introduced Talents, China (S022022045), the Key Program of Anhui Polytechnic University, China (No. KZ42022176).

**Institutional Review Board Statement:** Not applicable.

**Informed Consent Statement:** Not applicable.

**Data Availability Statement:** Not applicable.

**Conflicts of Interest:** The authors declare no conflict of interest.

## Appendix A

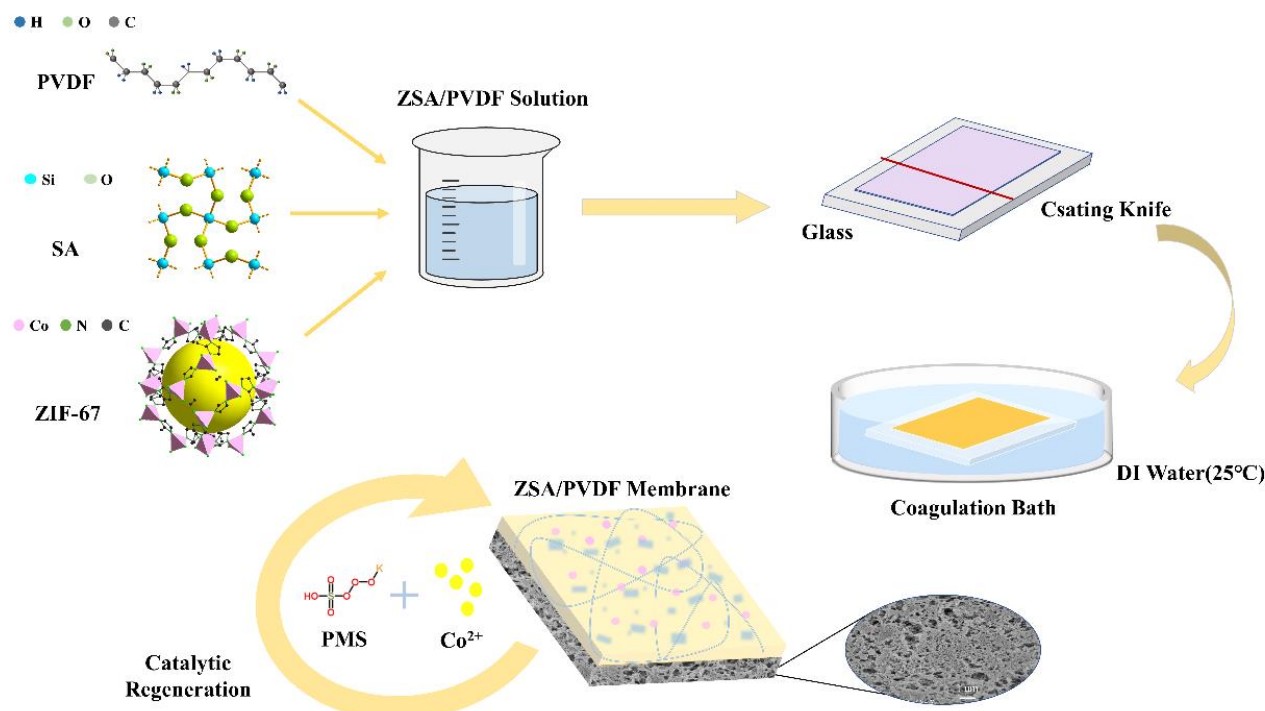

**Figure A1.** ZIF-67/SA@PVDF Ultrafiltration Membrane with Simultaneous Adsorption and Catalytic Oxidation for Dyes.

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
