# Peer review of "ZIF-67/SA@PVDF Ultrafiltration Membrane with Simultaneous Adsorption and Catalytic Oxidation for Dyes"

_sustainability, doi:10.3390/su15042879_

Round 1
Reviewer 1 Report
In this manuscript, a ZIF-67/SA@PVDF (ZSA3@PVDF) mixed matrix membrane has been fabricated by incorporating silicon aerogel (SA) and ZIF-67 nanoparticles in polyvinylidene fluoride (PVDF) membrane for the removal and degradation of dyes from water, which the water flux of ZSA3@PVDF membrane is 427.6 L m-2 h-1 bar-1 with the MB removal rate higher than 99% when filtrating 100 mL MB solution (5 mg/L). However, there are still some issues should be considered before publish.
1. The author should offer the preparation process of ZIF-67.
2. The samples is lack of measurement, and the BET test and FTIR test is necessary.
3. Some Figures should be improved the resolution and readability. All the axis scale should be enlarged, and the transverse longitudinal ratio of Figures should be coincident.
5. The language should be polished and the write-up should be further checked. For example, the title of introduction is missing, while the number of SiO2 in introduction should mark as subscript.
6. More related references about the MOFs should be referred (ex., https://doi.org/10.1002/smll.202203964, 10.1021/acsami.6b13153 and http://dx.doi.org/10.1126/science.1230444).
Reviewer 2 Report
This work presented in the manuscript entitled “ZIF-67/SA@PVDF ultrafiltration membrane with simultaneous adsorption and catalytic oxidation for dyes” is interesting and well presented. The authors have described the concept to a greater extent but the manuscript still needs some Minor corrections before publishing in Sustainability.
I appreciate the author's effort in this good study. However, the following comments need to be addressed.
Comment 1: Grammatical and formatting issues are so many there in the manuscript in several places, also check superscripts and subscripts errors. For. e.g. “1. Introduction” subtitle is missing in the manuscript; “SiO2” should be “SiO2”
Comment 2: In the abstract section, the authors should write about the techniques used to characterize the ZIF-67@PVDF and (Z)SAx@PVDF membranes.
Comment 3: Define the full form of “ZIF67”, “PMS”, and “MB” in the abstract section.
Comment 4: The introduction section needs to be improved. The authors should cite some relevant references in this section.
Ren, Yi, Ting Li, Weiming Zhang, Shu Wang, Mengqi Shi, Chao Shan, Wenbin Zhang, et al. "MIL-PVDF blend ultrafiltration membranes with ultrahigh MOF loading for simultaneous adsorption and catalytic oxidation of methylene blue." Journal of hazardous materials 365 (2019): 312-321.
Yin, Jiulong, Hai Tang, Di Liu, Tingting Huang, and Lei Zhu. "Application of ZIF-67 as a crosslinker to prepare sulfonated polysulfone mixed-matrix membranes for enhanced water permeability and separation properties." Water Science and Technology 84, no. 1 (2021): 144-158.
Cheng, Lilantian, Lei Li, Xiao Pei, Yun Ma, Fei Liu, and Jian Li. "PVDF/MOFs mixed matrix ultrafiltration membrane for efficient water treatment." Frontiers in Chemistry 10 (2022).
Comment 5: The manuscript Highlights: are mentioned in the manuscript according to the MDPI format there is no need for a Highlights section in the manuscript. So move the highlights section to the Supplementary file or delete it from the manuscript.
Comment 6: Include the purity of all chemicals used in the current study in section 2.1. Materials.
Comment 7: Include the Brunauer-Emmett-Teller (BET) instrument details in Section 2.3.
Comment 8: In section 2.4., the text mentioned equations (1) and (2): in the principle, they wrote different numbers for equations like (3) and (4). So correct it to (1) and (2). Also, correct the equation numbers according to the sequence order.
Comment 9: The authors mentioned this in section 3.1. mean particle size in the range of 300–600 nm. So, draw the particle size distribution graph for Figure 1(b).
Comment 10: Improve Figure 5 resolution.
Comment 11: Overall, the “Discussion” section needs more comparative study than just mentioning the obtained results. So please compare the obtained results with previous studies.
Comment 12: The conclusion needs to be revised properly.
Comment 13: The homogeneity of the reference section needs to be maintained. So please check and revise the references accordingly to the journal's instructions.
Round 2
Reviewer 1 Report
The authors have been answer the questions well, and I suggest the editors to publish this paper.